# Scalable inference of topic evolution via models for latent geometric structures

**Mikhail Yurochkin**
IBM Research
mikhail.yurochkin@ibm.com

**Zhiwei Fan**
University of Wisconsin-Madison
zhiwei@cs.wisc.edu

**Aritra Guha**
University of Michigan
aritra@umich.edu

**Paraschos Koutris**
University of Wisconsin-Madison
paris@cs.wisc.edu

**XuanLong Nguyen**
University of Michigan
xuanlong@umich.edu

## Abstract

We develop new models and algorithms for learning the temporal dynamics of the topic polytopes and related geometric objects that arise in topic model based inference. Our model is nonparametric Bayesian and the corresponding inference algorithm is able to discover new topics as the time progresses. By exploiting the connection between the modeling of topic polytope evolution, Beta-Bernoulli process and the Hungarian matching algorithm, our method is shown to be several orders of magnitude faster than existing topic modeling approaches, as demonstrated by experiments working with several million documents in under two dozens of minutes.[1]

## 1 Introduction

The topic or population polytope is a fundamental geometric object that underlies the presence of latent topic variables in topic and admixture models [4, 19, 21]. The geometry of topic models provides the theoretical basis for posterior contraction analysis of latent topics, in addition to helping to develop fast and quite accurate inference algorithms in parametric and nonparametric settings [18, 28, 29, 32]. When data and the associated topics are indexed by time dimension, it is of interest to study the temporal dynamics of such latent geometric structures. In this paper, we will study the modeling and algorithms for learning temporal dynamics of topic polytope that arises in the analysis of text corpora.

Several authors have extended the basic topic modeling framework to analyze how topics evolve over time. The Dynamic Topic Models (DTM) [3] demonstrated the importance of accounting for non-exchangeability between document groups, particularly when time index is provided. Another approach is to keep topics fixed and consider only evolving topic popularity [26]. Hong *et al.* [13] extended such an approach to multiple corpora. Ahmed and Xing [1] proposed a nonparametric construction extending DTM where topics can appear or eventually die out. Although the evolution of the latent geometric structure (i.e., the topic polytope) is implicitly present in these works, it was not explicitly addressed nor is the geometry exploited. A related limitation shared by these modeling

frameworks is the lack of scalability, due to inefficient joint modeling and learning of topics at each time point and topic evolution over time. To improve scalability, a natural solution is decoupling the two phases of inference.

To this end, we seek to develop a series of topic *meta*-models, i.e. models for temporal dynamics of topic polytopes, assuming that the topic estimates from each time point have already been obtained via some efficient static topic inference technique. The focus on inference of topic evolution offers novel opportunities and challenges. To start, what is the suitable ambient space in which the topic polytope is represented? As topics evolve, so are the number of topics that may become active or dormant, raising distinct modeling choices. Interesting issues arise in the inference, too. For instance, what is the principled way of *matching* vertices of a collection of polytopes to their next reincarnations? Such question arises because we consider modeling of topics learned independently across timestamps and text corpora, which entails the need for preserving the topic structure's permutation invariance of the vertex labels.

We consider an isometric embedding of the unit sphere in the word simplex, so that the evolution of topic polytopes may be represented by a collection of (random) trajectories of points residing on the unit sphere. Instead of attempting to mix-match vertices in an ad hoc fashion, we appeal to a Bayesian nonparametric modeling framework that allows the number of topic vertices to be random and vary across time. The mix-matching between topics shall be guided by the assumption on the smoothness of the collection of global trajectories on the sphere using von Mises-Fisher dynamics [15]. The selection of active topics at each time point will be enabled by a nonparametric prior on the random binary matrices via the (hierarchical) Beta-Bernoulli process [24].

Our contribution includes a sequence of Bayesian nonparametric models in increasing levels of complexity: the simpler model describes a topic polytope evolving over time, while the full model describes the temporal dynamics of a collection of topic polytopes as they arise from multiple corpora. The semantics of topics can be summarized as follows: there is a collection of latent global topics of unknown cardinality evolving over time (e.g. topics in science or social topics in Twitter). Each year (or day) a subset of the global topics is elucidated by the community (some topics may be dormant at a given time point). The nature of each global topic may change smoothly (via varying word frequencies). Additionally, different subsets of global topics are associated with different groups (e.g. journals or Twitter location stamps), some becoming active/inactive over time.

Another key contribution includes a suite of scalable approximate inference algorithms suitable for online and distributed settings. In particular, we focus mainly on MAP updates rather than a full Bayesian integration. This is appropriate in an online learning setting, moreover such updates of the latent topic polytope can be viewed as solving an optimal matching problem for which a fast Hungarian matching algorithm can be applied. Our approach is able to perform dynamic nonparametric topic inference on 3 million documents in 20 minutes, which is significantly faster than prior static online and/or distributed topic modeling algorithms [16, 12, 25, 6, 5].

The remainder of the paper is organized as follows. In Section 2 we define a Markov process over the space of topic polytopes (simplices). In Section 3 we present a series of models for polytope dynamics and describe our algorithms for online dynamic and/or distributed inference. Section 4 demonstrates experimental results. We conclude with a discussion in Section 5.

## 2 Temporal dynamics of a topic polytope

The fundamental object of inference in this work is the topic polytope arising in topic modeling which we shall now define [4, 18]. Given a vocabulary of $V$ words, a topic is defined as a probability distribution on the vocabulary. Thus a topic is taken to be a point in the vocabulary simplex, namely, $\Delta^{V-1}$, and a topic polytope for a corpus of documents is defined as a convex hull of topics associated with the documents. Geometrically, the topics correspond to the vertices (extreme points) of the (latent) topic polytope to be inferred from data.

In order to infer about the temporal dynamics of a topic polytope, one might consider the evolution of each topic variable, say $\theta^{(t)}$, which represents a vertex of the polytope at time $t$. A standard approach is due to Blei and Lafferty [3], who proposed to use a Gaussian Markov chain $\theta^{(t)}|\theta^{(t-1)} \sim \mathcal{N}(\theta^{(t-1)}, \sigma I)$ in $\mathbb{R}^V$ for modeling temporal dynamics and a logistic normal transfor-

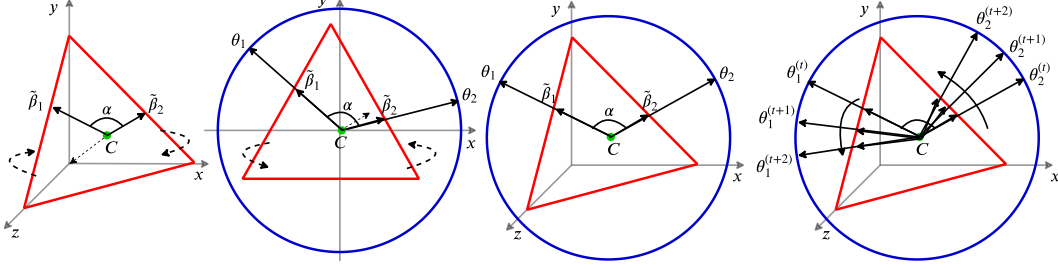

Figure 1: Invertible transformation between unit sphere and a standard simplex; dynamics example

mation $\pi(\theta^{(t)})_i := \frac{\exp(\theta_i^{(t)})}{\sum_i \exp(\theta_i^{(t)})}$, which sends elements of $\mathbb{R}^V$ into $\Delta^{V-1}$. In our meta-modeling approach, we consider topics, i.e. points in $\Delta^{V-1}$, learned independently across time and corpora. Logistic normal map is many-to-one, hence it is undesirably ambiguous in mapping a collection of topic polytopes to $\mathbb{R}^V$.

We propose to represent each topic variable as a point in a unit sphere $\mathbb{S}^{V-2}$, which possesses a natural isometric embedding (i.e. one-to-one) in the vocabulary simplex $\Delta^{V-1}$, so that the temporal dynamics of a topic variable can be identified as a (random) trajectory on $\mathbb{S}^{V-2}$. This trajectory shall be modeled as a Markovian process on $\mathbb{S}^{V-2}$: $\theta^{(t)}|\theta^{(t-1)} \sim \text{vMF}(\theta^{(t-1)}, \tau_0)$. Von Mises-Fisher (vMF) distribution is commonly used in the field of directional statistics [15] to model points on a unit sphere and was previously utilized for text modeling [2, 20].

**Isometric embedding of $\mathbb{S}^{V-2}$ into the vocabulary simplex** We start with the directional representation of topic polytope [29]: let $B = \{\beta_1, \ldots, \beta_K\}$ be a collection of vertices of a topic polytope. Each vertex is represented as $\beta_k := C + R_k\tilde{\beta}_k$, where $C \in \text{Conv}(B)$ is a reference point in a convex hull of $B$, $\tilde{\beta}_k \in \mathbb{R}^V$ is a topic direction and $R_k \in \mathbb{R}_+$. Moreover, $R_k \in [0, 1]$ is determined so that the tip of direction vector $\tilde{\beta}_k$ resides on the boundary of $\Delta^{V-1}$. Since the effective dimensionality of $\tilde{\beta}_k$ is $V - 2$, we can now define an one-to-one and isometric map sending $\tilde{\beta}_k$ onto $\mathbb{S}^{V-2}$ as follows: map of the vocabulary simplex $\Delta^{V-1} \in \mathbb{R}^V$ where it is first translated so that $C$ becomes the origin and then rotated into $\mathbb{R}^{V-1}$, where resulting topics, say $\theta_1, \ldots, \theta_K \in \mathbb{S}^{V-2}$, are normalized to the unit length. Observe that this geometric map is an isometry and hence invertible. It preserves angles between vectors, therefore we can evaluate vMF density without performing the map explicitly, by simply setting $\theta_k := \frac{\beta_k - C}{\|\beta_k - C\|}$. The following lemma formalizes this idea.

**Lemma 1.** $\Gamma : \{\beta = (\beta_1, \ldots, \beta_V) \in \Delta^{V-1} : \beta_i = 0 \text{ for some } i\} \to \{\theta \in \mathbb{S}^{V-1} : \mathbb{1}_V^T \theta = 0\}$ is a homeomorphism, where $\Gamma(\beta) = (\beta - C)/\|\beta - C\|_2$, and $\Gamma^{-1}(\theta) = -\frac{\theta}{\max_i \theta_i/c_i} + C$, for any $C = (c_1, \ldots, c_V) \in \Delta^{V-1}$.

Proofs of this Lemma and subsequent technical results are given in the Supplement. The intuition behind the construction is provided via Figure 1 which gives a geometric illustration for $V = 3$, vocabulary simplex $\Delta^{V-1}$ shown as red triangle. Two topics on the boundary (face) of the vocabulary simplex are $\beta_1 = C + \tilde{\beta}_1$ and $\beta_2 = C + \tilde{\beta}_2$. Green dot $C$ is the reference point and $\alpha = \angle(\tilde{\beta}_1, \tilde{\beta}_2)$. In Fig. 1 (left) we move $C$ by translation to the origin and rotate $\Delta^{V-1}$ from $xyz$ to $xy$ plane. In Fig. 1 (center left) we show the resulting image of $\Delta^{V-1}$ and add a unit sphere (blue) in $\mathbb{R}^2$. Corresponding to $\beta_1, \beta_2$ topics are the points $\theta_1, \theta_2$ on the sphere with $\angle(\theta_1, \theta_2) = \alpha$. Now, apply the inverse translation and rotation to *both* $\Delta^{V-1}$ and $\mathbb{S}^{V-2}$, the result is shown in Fig. 1 (center right) — we are back to $\mathbb{R}^3$ and $\angle(\theta_1, \theta_2) = \angle(\tilde{\beta}_1, \tilde{\beta}_2) = \alpha$, where $\theta_k = \frac{\beta_k - C}{\|\beta_k - C\|_2}$. In Fig. 1 (right) we give a geometric illustration of the temporal dynamics.

As described above, each topic evolves in a random trajectory residing in a unit sphere, so the evolution of a collection of topics can be modeled by a collection of corresponding trajectories on the sphere. Note that the number of "active" topics may be unknown and vary over time. Moreover, a topic may be activated, become dormant, and then resurface after some time. New modeling elements are introduced in the next section to account for these phenomena.

# 3 Hierarchical Bayesian modeling for single or multiple topic polytopes

We shall present a sequence of models with increasing levels of complexity: we start by introducing a hierarchical model for online learning of the temporal dynamics of a single topic polytope, allowing for varying number of vertices over time. Next, a static model for *multiple* topic polytopes learned on different corpora drawing on a common pool of global topics. Finally, we present a "full" model for modeling evolution of global topic trajectories over time and across groups of corpora.

## 3.1 Dynamic model for single topic polytope

At a high level, our model maintains a collection of global trajectories taking values on a unit sphere. Each trajectory shall be endowed with a von Mises-Fisher dynamic described in the previous section. At each time point, a random topic polytope is constructed by selecting a (random) subset of points on the trajectory evaluated at time $t$. The random selection is guided by a Beta-Bernoulli process prior [24]. This construction is motivated by a modeling technique of Nguyen [17], who studied a Bayesian hierarchical model for inference of smooth trajectories on an Euclidean domain using Dirichlet process priors. Our generative model, using Beta-Bernoulli process as a building block, is more appropriate for the purpose of topic discovery. Due to the isometric embedding of $\mathbb{S}^{V-2}$ in $\Delta^{V-1}$ described in the previous section, from here on we shall refer to topics as points on $\mathbb{S}^{V-2}$.

First, generate a collection of global topic trajectories using Beta Process prior (cf. Thibaux and Jordan [24]) [2] with a base measure $H$ on the space of trajectories on $\mathbb{S}^{V-2}$ and mass parameter $\gamma_0$:

$$Q|\gamma_0, H \sim \text{BP}(\gamma_0, H). \tag{1}$$

It follows that $Q = \sum_i q_i \delta_{\theta_i}$, where $\{q_i\}_{i=1}^{\infty}$ follows a stick-breaking construction [23]: $\mu_i \sim \text{Beta}(\gamma_0, 1)$, $q_i = \prod_{j=1}^{i} \mu_j$, and each $\theta_i \sim H$ is a sequence of $T$ random elements on the unit sphere $\theta_i := \{\theta_i^{(t)}\}_{t=1}^{T}$, which are generated as follows:

$$
\begin{aligned}
&\theta_i^{(t)}|\theta_i^{(t-1)} \sim \text{vMF}(\theta_i^{(t-1)}, \tau_0) \text{ for } t = 1, \dots, T, \\
&\theta_i^{(0)} \sim \text{vMF}(\cdot, 0) - \text{uniform on } \mathbb{S}^{V-2}.
\end{aligned}
\tag{2}
$$

At any given time $t = 1, \dots, T$, the process $Q$ induces a marginal measure $Q_t$, whose support is given by the atoms of $Q$ as they are evaluated at time $t$. Now, select a subset of the global topics that are active at $t$ via the Bernoulli process $\mathcal{T}^{(t)}|Q_t \sim \text{BeP}(Q_t)$. Then $\mathcal{T}^{(t)} := \sum_{i=1} b_i^{(t)} \delta_{\theta_i^{(t)}}$, where $b_i^{(t)}|q_i \sim \text{Bern}(q_i), \forall i$. $\mathcal{T}^{(t)}$ are supported by atoms $\{\theta_i^{(t)} : b_i^{(t)} = 1, i = 1, 2, \dots\}$ representing topics active at time $t$. Finally, assume that noisy measurements of each of these topic variables are generated via:

$$
\begin{aligned}
&v_k^{(t)}|\mathcal{T}^{(t)} \sim \text{vMF}(\mathcal{T}_k^{(t)}, \tau_1), \ k = 1, \dots, K^{(t)}, \text{ where} \\
&K^{(t)} := \text{card}(\mathcal{T}^{(t)}); \mathcal{T}_k^{(t)} \text{ is } k\text{-th atom of } \mathcal{T}^{(t)}.
\end{aligned}
\tag{3}
$$

Noisy estimates for the topics at any particular time point may come from either the global topics observed until the previous time point or a topic yet unexplored. We emphasize that topics $\{v_k^{(t)}\}_{k=1}^{K^{(t)}}$ for $t = 1, \dots, T$ are the quantities we aim to model, hence we refer to our approach as the *meta*-model. These topics may be learned, for each time point independently, by any stationary topic modeling algorithms, and then transformed to sphere by applying Lemma 1.

Let $B^{(t)}$ denote the binary matrix representing the assignment of observed topic estimates to global topics at time point $t$, i.e, $B_{ik}^{(t)} = 1$ if the vector $v_k^{(t)}$ is a noisy estimate for $\theta_i^{(t)}$. In words, these random variables "link up" the noisy estimates at any time point to the global topics observed thus far. By conditional independence, the joint posterior of the hidden $\theta^{(t)}$ given observed noisy $v^{(t)}$ is:

$$\mathbb{P}\left(\theta^{(0)}, \{\theta^{(t)}, B^{(t)}\}_{t=1}^{T}|\{v^{(t)}\}_{t=1}^{T}\right) \propto \mathbb{P}(\theta^{(0)}) \prod_{t=1}^{T} \mathbb{P}(\theta^{(t)}, B^{(t)}|\theta^{(t-1)}, \{B^{(a)}\}_{a=1}^{t-1})\mathbb{P}(v^{(t)}|\theta^{(t)}, B^{(t)}).$$

At $t$, $\mathbb{P}(\theta^{(t)}, B^{(t)}|\theta^{(t-1)}, \{B^{(a)}\}_{a=1}^{t-1})\mathbb{P}(v^{(t)}|\theta^{(t)}, B^{(t)}) \propto \mathbb{P}(\theta^{(t)}, B^{(t)}|\theta^{(t-1)}, v^{(t)}, \{B^{(a)}\}_{a=1}^{t-1}) \propto$

$$\prod_{i=1}^{L_{t-1}} \left( \left( m_i^{(t-1)}/(t-m_i^{(t-1)}) \right)^{\sum_{k=1}^{K^{(t)}} B_{ik}^{(t)}} \exp(\tau_0 \langle \theta_i^{(t-1)}, \theta_i^{(t)} \rangle) \right)$$
$$\cdot \frac{\exp(-\frac{\gamma_0}{t})(\gamma_0/t)^{L_t - L_{t-1}}}{(L_t - L_{t-1})!} \exp(\tau_1 \sum_{i=1}^{L_t} \sum_{k=1}^{K^{(t)}} B_{ik}^{(t)} \langle \theta_i^{(t)}, v_k^{(t)} \rangle). \tag{4}$$

The equation above represents a product of four quantities: (1) probability of $B^{(t)}$s, where $m_i^{(t)}$ denotes the number of occurrences of topic $i$ up to time point $t$ (cf. popularity of a dish in the Indian Buffet Process (IBP) metaphor [9]), (2) vMF conditional of $\theta_i^{(t)}$ given $\theta_i^{(t-1)}$ (cf. Eq. (2)), (3) number of new global topics at time $t$, $L_t - L_{t-1} \sim \text{Pois}(\gamma_0/t)$, and (4) emission probability $\mathbb{P}(v^{(t)}|\theta^{(t)}, B^{(t)})$ (cf. Eq. (3)). Derivation details are given in the Supplement.

**Streaming Dynamic Matching (SDM)**　To perform MAP estimation in the streaming setting, we highlight the connection of the maximization of the posterior (4) to the objective of an optimal *matching* problem: given a cost matrix, workers should be assigned to tasks, at most one worker per task and one task per worker. The solution of this problem is obtained by employing the well-known Hungarian algorithm [14]. In the context of dynamic topic modeling, our goal is to match topics learned on the new timestamp to the trajectories of topics learned over the previous timestamps, where the cost is governed by our model. This connection is formalized by the following.

**Proposition 1.** Given the cost $C_{ik}^{(t)} = \begin{cases} \|\tau_1 v_k^{(t)} + \tau_0 \theta_i^{(t-1)}\|_2 - \tau_0 + \log \frac{m_i^{(t-1)}}{t-m_i^{(t-1)}}, i \le L_{t-1} \\ \tau_1 + \log \frac{\gamma_0}{t} - \log(i - L_{t-1}), L_{t-1} < i \le L_{t-1} + K^{(t)} \end{cases}$

consider the optimization problem $\max_{B^{(t)}} \sum_{i,k} B_{ik}^{(t)} C_{ik}^{(t)}$ subject to the constraints that (a) for each fixed $i$, at most one of $B_{ik}^{(t)}$ is 1 and the rest are 0, and (b) for each fixed $k$, exactly one of $B_{ik}^{(t)}$ is 1 and the rest are 0. Then, the MAP estimate for Eq. (4) can be obtained by the Hungarian algorithm, which solves for $((B_{ik}^{(t)}))$ to obtain $\theta_i^{(t)}$ as

$$\begin{cases} \frac{\tau_1 v_k^{(t)} + \tau_0 \theta_i^{(t-1)}}{\|\tau_1 v_k^{(t)} + \tau_0 \theta_i^{(t-1)}\|_2}, & \text{if } \exists k \text{ s.t. } B_{ik}^{(t)} = 1 \text{ and } i \le L_{t-1} \\ v_k^{(t)}, & \text{if } \exists k \text{ s.t. } B_{ik}^{(t)} = 1 \text{ and } i > L_{t-1} \text{ (new topic)} \\ \theta_i^{(t-1)} & \text{otherwise (topic is dormant at } t). \end{cases} \tag{5}$$

We defer proof to the Supplement. To complete description of the inference we shall discuss how noisy estimates are obtained from the bag-of-words representation of the documents observed at time point $t$. We choose to use CoSAC [29] algorithm to obtain topics $\{\beta_k^{(t)} \in \Delta^{V-1}\}_{k=1}^{K^{(t)}}$ from $\{x_m^{(t)} \in \mathbb{N}^V\}_{m=1}^{M_t}$, collection of $M_t$ documents at time point $t$. CoSAC is a stationary topic modeling algorithm which can infer number of topics from the data and is computationally efficient for moderately sized corpora. We note that other topic modeling algorithms, e.g., variational inference [4] or Gibbs sampling [11, 22], can be used in place of CoSAC. Estimated topics are then transformed to $\{v_k^{(t)} \in \mathbb{S}^{V-2}\}_{k=1}^{K^{(t)}}$ using Lemma 1 and reference point $C_t = \sum_{a=1}^{t} \sum_{m=1}^{M_a} \frac{x_m^{(a)}}{N_m^{(a)}} / \sum_{a=1}^{t} M_a$, where $N_m^{(a)}$ is the number of words in the corresponding document. Our reference point is simply an average (computed dynamically) of the normalized documents observed thus far. Finally we update MAP estimates of global topics dynamics based on Proposition 1. Streaming Dynamic Matching (SDM) is summarized in Algorithm 1.

**Additional related literature**　utilizing similar technical building blocks in different contexts. Fox *et al.* [8] utilized Beta-Bernoulli process in time series modeling to capture switching regimes of an autoregressive process, where the corresponding Indian Buffet Process was used to select subsets of the latent states of the Hidden Markov Model. Williamson *et al.* [27] used Indian Buffet Process in topic models to sparsify document topic proportions. Campbell *et al.* [7] utilized Hungarian algorithm for streaming mean-field variational inference of the Dirichlet Process mixture model.

---

**Algorithm 1** Streaming Dynamic Matching (SDM)

---

1: **for** $t = 1, \ldots, T$ **do**
2:      Observe documents $\{x_m^{(t)}\}_{m=1}^{M_t}$
3:      Estimate topics $\{\beta_k^{(t)}\}_{k=1}^{K^{(t)}} = \text{CoSAC}(\{x_m^{(t)}\}_{m=1}^{M_t})$
4:      Map topics to sphere $\{v_k^{(t)}\}_{k=1}^{K^{(t)}}$ (Lemma 1)
5:      Given $\{\theta^{(t-1)}\}_{i=1}^{L_{t-1}}$ and $\{v_k^{(t)}\}_{k=1}^{K^{(t)}}$ compute cost matrix as in Proposition 1
6:      Using Hungarian algorithm solve the corresponding matching problem to obtain $B^{(t)}$
7:      Compute $\{\theta^{(t)}\}_{i=1}^{L_t}$ as in eq. (5)

---

## 3.2    Beta-Bernoulli Process for multiple topic polytopes

We now consider meta-modeling in the presence of multiple corpora, each of which maintains its own topic polytope. Large text corpora often can be partitioned based on some grouping criteria, e.g. scientific papers by journals, news by different media agencies or tweets by location stamps. In this subsection we model the collection of topic polytopes observed at a single time point by employing the Beta-Bernoulli Process prior [24]. The modeling of a collection of polytopes evolving over time will be described in the following subsection.

First, generate global topic measure $Q$ as in Eq. (1). Here, we are interested only in a single time point, the base measure $H$ is simply a $\text{vMF}(\cdot, 0)$, the uniform distribution over $\mathbb{S}^{V-2}$. Next, for each group $j = 1, \ldots, J$, select a subset of the global topics:

$$\mathcal{T}_j | Q \sim \text{BeP}(Q), \text{ then } \mathcal{T}_j := \sum_{i=1} b_{ji} \delta_{\theta_i}, \text{ where } b_{ji}|q_i \sim \text{Bern}(q_i), \forall i. \tag{6}$$

Notice that each group $\mathcal{T}_j := \{\theta_i : b_{ji} = 1, i = 1, 2, \ldots\}$ selects only a subset from the collection of global topics, which is consistent with the idea of partitioning by journals: some topics of ICML are not represented in SIGGRAPH and vice versa. The next step is analogous to Eq. (3):

$$v_{jk}|\mathcal{T}_j \sim \text{vMF}(\mathcal{T}_{jk}, \tau_1) \text{ for } k = 1, \ldots, K_j, \text{ where } K_j := \text{card}(\mathcal{T}_j). \tag{7}$$

We again use $B$ to denote the binary matrix representing the assignment of global topics to the noisy topic estimates, i.e., $B_{jik} = 1$ if the $k^{th}$ topic estimate for group $j$ arises as a noisy estimate of global topic $\theta_i$. However, the *matching* problem is now different from before: we don't have any information about the global topics as there is no history, instead we should match a *collection* of topic polytopes to a global topic polytope. The matrix of topic assignments is distributed a priori by an Indian Buffet Process (IBP) with parameter $\gamma_0$. The conditional probability for global topics $\theta_i$ and assignment matrix $B$ given topic estimates $v_{jk}$ has the following form:

$$\mathbb{P}(B, \theta|v) \propto \exp(\tau_1 \textstyle\sum_{j,i,k} B_{jik}\langle\theta_i, v_{jk}\rangle)\text{IBP}(\{m_i\}), \text{ where } m_i = \textstyle\sum_{j,k} B_{jik} \tag{8}$$

and IBP is the prior (see Eq. (15) in [10]) with $m_i$ denoting the popularity of global topic $i$.

**Distributed Matching (DM)**    Similar to Section 3.1, we look for point estimates for the topic directions $\theta$ and for the topic assignment matrix $B$. Direct computation of the global MAP estimate for Eq. (8) is not straight-forward. The problem of matching across groups and topics is not amenable to a closed form Hungarian algorithm. However we show that for a fixed group the assignment optimization reduces to a case of the Hungarian algorithm. This motivates the use of Hungarian algorithm iteratively, which guarantees convergence to a local optimum.

**Proposition 2.** Given the cost

$$C_{jik} = \begin{cases} \tau_1\|v_{jk} + \sum_{-j,i,k} B_{-jik}v_{-jk}\|_2 - \tau_1\|\sum_{-j,i,k} B_{-jik}v_{-jk}\|_2 + \log\frac{m_{-ji}}{J-m_{-ji}}, \text{ if } i \leq L_{-j} \\ \tau_1 + \log\frac{\gamma_0}{J} - \log(i - L_{-j}), \text{ if } L_{-j} < i \leq L_{-j} + K_j, \end{cases}$$

where $-j$ denotes groups excluding group $j$ and $L_{-j}$ is the number of global topics before group $j$ (due to exchangeability of the IBP, group $j$ can always be considered last). Then, a locally optimum MAP estimate for Eq. (8) can be obtained by iteratively employing the Hungarian algorithm to solve: for each group $j$, $(((B_{jik})))$ which maximizes $\sum_{j,i,k} B_{jik}C_{jik}$, subject to constraints: (a) for each fixed $i$ and $j$, at most one of $B_{jik}$ is 1, rest are 0 and (b) for each fixed $k$ and $j$, exactly one of $B_{jik}$ is 1, rest are 0. After solving for $(((B_{jik})))$, $\theta_i$ is obtained as $\theta_i = \frac{\sum_{j,k} B_{jik}v_{jk}}{\|\sum_{j,k} B_{jik}v_{jk}\|_2}$.

The noisy topics for each of the groups can be obtained by applying CoSAC to corresponding documents, which is trivially parallel. Distributed Matching algorithm and proof of the Proposition 2 are given in the Supplement.

### 3.3 Dynamic Hierarchical Beta Process

Our "full" model, the Dynamic Hierarchical Beta Process model (dHBP), builds on the constructions described in subsections 3.1 and 3.2 to enable the inference of temporal dynamics of collections of topic polytopes. We start by specifying the upper level Beta Process given by Eq. (1) and base measure $H$ given by Eq. (2). Next, for each group $j = 1, \ldots, J$, we introduce an additional level of hierarchy to model group specific distributions over topics

$$Q_j | Q \sim \mathrm{BP}(\gamma_j, Q), \text{ then } Q_j := \sum_i p_{ji} \delta_{\theta_i}, \tag{9}$$

where $p_{ji}$s vary around corresponding $q_i$. The distributional properties of $p_{ji}$ are described in [24].

At any given time $t$, each group $j$ selects a subset from the common pool of global topics:

$$\mathcal{T}_j^{(t)} | Q_{jt} \sim \mathrm{BeP}(Q_{jt}), \text{ then } \mathcal{T}_j^{(t)} := \sum_{i=1} b_{ji}^{(t)} \delta_{\theta_i^{(t)}}, \text{ where } b_{ji}^{(t)} | p_{ji} \sim \mathrm{Bern}(p_{ji}), \forall i. \tag{10}$$

Let $\mathcal{T}_j^{(t)} := \{\theta_i^{(t)} : b_{ji}^{(t)} = 1, i = 1, 2, \ldots\}$ be the corresponding collection of atoms – topics active at time $t$ in group $j$. Noisy measurements of these topics are generated by:

$$v_{jk}^{(t)} | \mathcal{T}_j^{(t)} \sim \mathrm{vMF}(\mathcal{T}_{jk}^{(t)}, \tau_1) \text{ for } k = 1, \ldots, K_j^{(t)}, \text{ where } K_j^{(t)} := \mathrm{card}(\mathcal{T}_j^{(t)}). \tag{11}$$

The conditional distribution of global topics at $t$ given the state of the global topics at $t - 1$ is

$$\mathbb{P}(\theta^{(t)}, B^{(t)} | \theta^{(t-1)}, v^{(t)}, \{B^{(a)}\}_{a=1}^{t-1}) \propto$$
$$\exp\left(\tau_0 \sum_{i=1} \langle \theta_i^{(t)}, \theta_i^{(t-1)} \rangle \right) F(\{m_{ji}^{(t-1)}\}, \{m_{ji}^{(t)}\}) \cdot \exp\left(\sum_{j,i,k} \tau_1 B_{jik}^{(t)} \langle \theta_i^{(t)}, v_{jk}^{(t)} \rangle \right), \tag{12}$$

where $F(\{m_{ji}^{(t-1)}\}, \{m_{ji}^{(t)}\})$ is the prior term dependent on the popularity counts history from current and previous time points. Analogous to the Chinese Restaurant Franchise [22], one can think of an Indian Buffet Franchise in the case of HBP. A headquarter buffet provides some dishes each day and the local branches serve a subset of those dishes. Although this analogy seems intuitive, we are not aware of a corresponding Gibbs sampler and it remains to be a question of future studies. Therefore, unfortunately, we are unable to handle this prior term directly and instead propose a heuristic replacement — stripping away popularity of topics across groups and only considering group specific topic popularity (groups still remain dependent through the atom locations).

**Streaming Dynamic Distributed Matching (SDDM)** We combine our results to perform approximate inference of the model in Section 3.3. Using Hungarian algorithm, iterating over groups at time $t$ obtain estimates for $(((B_{jik}^{(t)})))$ based on the following cost $C_{jik}^{(t)} =$

$$\begin{cases} \|\tau_1 v_{jk}^{(t)} + \tau_1 \sum_{-j,i,k} B_{-jik}^{(t)} v_{-jk}^{(t)} + \tau_0 \theta_i^{(t-1)}\|_2 - \|\sum_{-j,i,k} B_{-jik}^{(t)} v_{-jk}^{(t)} + \tau_0 \theta_i^{(t-1)}\|_2 + \log \frac{1+m_{ji}^{(t)}}{t-m_{ji}^{(t)}}, \\ \tau_1 + \log \frac{\gamma_0}{J} - \log(i - L_{-j}^{(t)}), \text{ if } L_{-j}^{(t)} < i \le L_{-j}^{(t)} + K_j^{(t)}, \end{cases}$$

where first case is if $i \le L_{-j}^{(t)}$; $m_{ji}^{(t)}$ denotes the popularity of topic $i$ in group $j$ up to time $t$ (plus one is used to indicate that global topic $i$ exists even when $m_{ji}^{(t)} = 0$). Then compute global topic estimates $\theta_i^{(t)} = \frac{\tau_1 \sum_{j,k} B_{jik}^{(t)} v_{jk}^{(t)} + \tau_0 \theta_i^{(t-1)}}{\|\tau_1 \sum_{j,k} B_{jik}^{(t)} v_{jk}^{(t)} + \tau_0 \theta_i^{(t-1)}\|_2}$. At time point $t$, the noisy topics for each of the groups can be obtained by applying CoSAC to corresponding documents in parallel. SDDM algorithm and cost derivations are presented in the Supplement.

## 4 Experiments

We study ability of our models to learn the latent temporal dynamics and discover new topics that change over time. Next we show that our models scale well by utilizing temporal and group inherent data structures. We also study hyperparameters choices. We analyze two datasets: the Early Journal Content (http://www.jstor.org/dfr/about/sample-datasets), and a collection of Wikipedia articles partitioned by categories and in time according to their popularity.

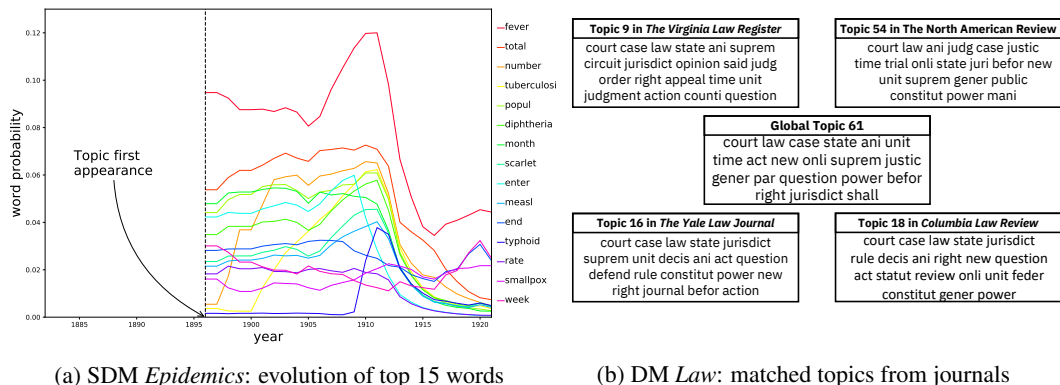

(a) SDM *Epidemics*: evolution of top 15 words      (b) DM *Law*: matched topics from journals

Figure 2: Qualitative examples of topics learned by SDM and DM algorithms on the EJC data

## 4.1 Temporal Dynamics and Topic Discovery

**Early Journal Content.** The Early Journal Content dataset spans years from $1665$ up to $1922$. Years before $1882$ contain very few articles, and we aggregated them into a single timepoint. After preprocessing, dataset has $400k$ scientific articles from over $400$ unique journals. The vocabulary was truncated to $4516$ words. We set all articles from the last available year ($1922$) aside for the testing purposes.

**Case study: epidemics.** The beginning of the 20th century is known to have a vast history of disease epidemics of various kinds, such as smallpox, typhoid, yellow fever to name a few. Vaccines or effective treatments for the majority of them were developed shortly after. One of the journals represented in the EJC dataset is the "Public Health Report"; however, publications from it are only available starting $1896$. Primary objective of the journal was to reflect epidemic disease infections. As one of the goals of our modeling approach is topic discovery, we verify that the model can discover an epidemics-related topic around $1896$. Figure 2a shows that SDM correctly discovered a new topic is $1896$ semantically related to epidemics. We plot the evolution of probabilities of the top $15$ words in this topic across time. We observe that word "typhoid" increases in probability towards $1910$ in the "epidemics" topic, which aligns with historical events such as Typhoid Mary in $1907$ and chlorination of public drinking water in the US in $1908$ for controlling the typhoid fever. The probability of "tuberculosis" also increases, aligning with foundation of the National Association for the Study and Prevention of Tuberculosis in $1904$.

**Case study: law.** Some of the EJC journals are related to the topic of law. Our DM algorithm identified a global topic semantically similar to law by matching similar topics present in 32 out of the 417 journals. In Figure 2b we present the learned global topic and 4 examples of the matched local topics with the corresponding journal names. Our algorithm correctly identified that these journals have a shared law topic.

## 4.2 Scalability

**Wiki Corpus.** We collected articles from Wikipedia and their page view counts for the 12 months of 2017 and category information (e.g., Arts, History). We used categories as groups and partitioned the data across time according

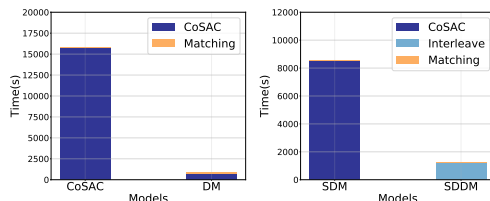

Figure 3: Comparison on Wiki Data (20 cores)

to the page view counts. Dataset construction details are given in the Supplement. The total number of documents is about 3 million, and we reduced vocabulary to 7359 words similarly to [12]. For testing we set aside documents from category Art from December 2017.

**Modeling Grouping.** In Fig. 3 we present comparisons on Wiki data: CoSAC [29] v.s DM under the static *distributed* setting and SDM v.s SDDM under the dynamic *streaming* setting. Fig. 3 (left) shows that for data accessible in groups, DM outperforms CoSAC by $\sim 25X$, as DM runs CoSAC on different data groups in parallel and then matches the outputs. Matching time adds only a small

Table 1: Modeling topics of EJC ‖ Modeling Wikipedia articles

|  | Perplexity | Time | Topics | Cores ‖ | Perplexity | Time | Topics | Cores |
|---|---|---|---|---|---|---|---|---|
| SDM | **1179** | 22min | 125 | 1 | 1254 | 2.4hours | 182 | 1 |
| DM | 1361 | 5min | 125 | 20 | 1260 | **15min** | 182 | 20 |
| SDDM | 1241 | **2.3min** | 103 | 20 | **1201** | 20min | 238 | 20 |
| DTM | 1194 | 56hours | 100 | 1 | NA | >72hours | 100 | 1 |
| SVB | 1840 | 3hours | 100 | 20 | 1219 | 29.5hours | 100 | 20 |
| CoSAC | 1191 | 51min | 132 | 1 | 1227 | 4.4hours | 173 | 1 |

overhead compared to the runtime of CoSAC. Similarly, in Fig. 3 (right), SDDM is $\sim 6X$ faster than SDM, since SDDM can process documents of different groups in parallel and interleaves CoSAC with matching: while matching is being performed on data groups with timestamp $t$, CoSAC can process the data that arrives with timestamp $t + 1$ in parallel.

**Modeling temporality** also benefits scalability. We compare our methods with other topic models on both Wiki and EJC datasets: Streaming Variational Bayes (SVB) [5] and Dynamic Topic Models (DTM) [3] trained with 100 topics. Perplexity scores on the held out data, training times, computing resources and number of topics are reported in Table 1. On the wiki dataset, SDDM took only 20min to process approximately 3 million documents, which is much faster than the other approaches.

Regarding perplexity scores, SDDM generally outperforms DM, which suggests that modeling time is beneficial. For the EJC dataset, SDM outperforms SDDM. Modeling groups might negatively affect perplexity because the majority of the EJC journals (groups) have very few articles (i.e. less than $100$ – a setup challenging for many topic modeling algorithms). On the Wiki corpus each category (group) has sufficient amount of training documents and time-group partitioning considered by SDDM achieves the best perplexity score.

### 4.3 Parameter choices

The rate of topic dynamics of the SDM and SDDM is effectively controlled by $\tau_0$, where smaller values imply higher dynamics rate. Parameter $\tau_1$ controls variance of local topics around corresponding global topics in all of our models. This variance dictates how likely a local topic to be matched to an existing global topic. When this variance is small, the model will tend to identify local topics as new global topics more often. Lastly, $\gamma_0$ affects the probability of new topic discovery, which scales with time and number of groups. In the preceding experiments we set $\tau_0 = 2, \tau_1 = 1, \gamma_0 = 1$ for SDM; $\tau_1 = 2, \gamma_0 = 1$ for DM; $\tau_0 = 4, \tau_1 = 2, \gamma_0 = 2$ for SDDM. In Figure 4 we show heatpmaps for perplexity and number of learned topics, fixing $\gamma_0 = 1$ and varying $\tau_0$ and $\tau_1$. We see that for large $\tau_1$, SDM identifies more topics to fit the smaller variability constraint imposed by the parameter.

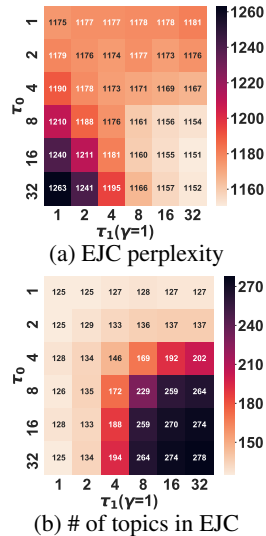

(a) EJC perplexity

(b) # of topics in EJC

Fig. 4: SDM parameters

## 5 Discussion and Conclusion

Our work suggests the naturalness of incorporating sophisticated Bayesian nonparametric techniques in the inference of rich latent geometric structures of interest. We demonstrated the feasibility of *approximate* nonparametric learning at scale, by utilizing suitable geometric representations and devising fast algorithms for obtaining reasonable point estimates for such representations. Further directions include incorporating more meaningful geometric features into the models (e.g., via more elaborated base measure modeling for the Beta Process) and developing efficient algorithms for full Bayesian inference. For instance, the latent geometric structure of the problem is solely encoded in the base measure. We want to explore choices of base measures for other geometric structures such as collections of k-means centroids, principal components, etc. Once an appropriate base measure is constructed, our Beta process based models can be utilized to enable a new class of Bayesian nonparametric models amenable to scalable inference and suitable for analysis of large datasets. In our concurrent work we have utilized model construction similar to one from Section 3.2 to perform Federated Learning of neural networks trained on heterogeneous data [31] and proposed a general framework for model fusion [30].

**Acknowledgments**

This research is supported in part by grants NSF CAREER DMS-1351362, NSF CNS-1409303, a research gift from Adobe Research and a Margaret and Herman Sokol Faculty Award to XN.

## Footnotes

[1]Code: https://github.com/moonfolk/SDDM

[2]Thibaux and Jordan [24] write $\text{BP}(c, H)$, $H(\Omega) = \gamma_0$; we set $c = 1$, $H = H/\gamma_0$ and write $\text{BP}(\gamma_0, H)$.

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
