[Supplementary Material]

# Supplementary material for Scalable inference of topic evolution via models for latent geometric structures

**Mikhail Yurochkin**
IBM Research
mikhail.yurochkin@ibm.com

**Zhiwei Fan**
University of Wisconsin-Madison
zhiwei@cs.wisc.edu

**Aritra Guha**
University of Michigan
aritra@umich.edu

**Paraschos Koutris**
University of Wisconsin-Madison
paris@cs.wisc.edu

**XuanLong Nguyen**
University of Michigan
xuanlong@umich.edu

## 1 Derivations of posterior probabilities

### 1.1 Dynamic Beta Process posterior

The departing point for arriving at MAP estimation algorithm for the Dynamic Beta Process proposed in Section 3.1 of the main text is the posterior derivation at a time point $t$ (Eq. (4) of the main text):

$$
\mathbb{P}(\theta^{(t)}, B^{(t)} | \theta^{(t-1)}, \{B^{(a)}\}_{a=1}^{t-1}) \mathbb{P}(v^{(t)} | \theta^{(t)}, B^{(t)}) \propto
$$
$$
\mathbb{P}(\theta^{(t)}, B^{(t)} | \theta^{(t-1)}, v^{(t)}, \{B^{(a)}\}_{a=1}^{t-1}) \propto
$$
$$
\prod_{i=1}^{L_{t-1}} \left( \left( \frac{m_i^{(t-1)}}{t - m_i^{(t-1)}} \right)^{\sum_{k=1}^{K^{(t)}} B_{ik}^{(t)}} \exp(\tau_0 \langle \theta_i^{(t-1)}, \theta_i^{(t)} \rangle) \right) \tag{1}
$$
$$
\cdot \frac{\exp(-\frac{\gamma_0}{t})(\gamma_0/t)^{L_t - L_{t-1}}}{(L_t - L_{t-1})!} \exp(\tau_1 \sum_{i=1}^{L_t} \sum_{k=1}^{K^{(t)}} B_{ik}^{(t)} \langle \theta_i^{(t)}, v_k^{(t)} \rangle).
$$

Starting with the vMF emission probabilities,

$$
\mathbb{P}(v^{(t)} | \theta^{(t)}, B^{(t)}) \propto \exp(\tau_1 \sum_{i=1}^{L_t} \sum_{k=1}^{K^{(t)}} B_{ik}^{(t)} \langle \theta_i^{(t)}, v_k^{(t)} \rangle), \tag{2}
$$

we obtain the last term of Eq. (1). The conditional distribution of $\theta^{(t)}, B^{(t)}$ given $\theta^{(t-1)}, \{B^{(a)}\}_{a=1}^{t-1}$, obtained when random variables $\{q_i\}_{i=1}^{\infty}$ are marginalized out, can be decomposed into two parts: parametric part – time $t$-th incarnations of subset of previously observed global topics $\theta^{(t-1)}$ and nonparametric part – number of new topics appearing at time $t$. The middle term can be seen to come from the Poisson prior on the number of new topics induced by the Indian Buffet Process (see [4] for details):

$$
L_t - L_{t-1} \sim \text{Pois}(\gamma_0/t)
$$
$$
\mathbb{P}(L_t - L_{t-1} = l_t - l_{t-1}) = \frac{\exp(-\frac{\gamma_0}{t})(\gamma_0/t)^{l_t - l_{t-1}}}{(l_t - l_{t-1})!}. \tag{3}
$$

Finally, the first term of Eq. (1) is composed of a probability of previously observed global topic to appear at time $t$:

$$\mathbb{P}(\sum_k B_{ik}^{(t)}, i \in \{1,\ldots,L_{t-1}\}|\{B^{(a)}\}_{a=1}^{t-1}) \propto (m_i^{(t-1)})^{\sum_k B_{ik}^{(t)}}(t - m_i^{(t-1)})^{1-\sum_k B_{ik}^{(t)}}, \quad (4)$$

where $m_i^{(t-1)}$ denotes the number of times topic $i$ appeared up to time $t$. Also, the base measure probability of the vMF dynamics is:

$$\mathbb{P}(\theta_i^{(t)}|\theta_i^{(t-1)}) \propto \exp(\tau_0 \langle \theta_i^{(t-1)}, \theta_i^{(t)} \rangle). \quad (5)$$

Combining Equations (2)–(5) we arrive at Eq. (1) (Eq. (4) of the main text).

## 1.2   Posterior of the Beta process for multiple topic polytopes

First recall Eq. (8) of the main text:

$$\mathbb{P}(B,\theta|v) \propto \exp(\tau_1 \sum_{j,i,k} B_{jik}\langle \theta_i, v_{jk} \rangle)\mathrm{IBP}(\{m_i\}), \quad (6)$$

To arrive at this result first note that $\mathbb{P}(B,\theta|v) \propto \mathbb{P}(v|\theta,B)\mathbb{P}(B)\mathbb{P}(\theta)$, where $\mathbb{P}(\theta)$ is a uniform distribution on sphere from the model specification of Section 3.2 of the main text and hence is a constant. Next, the likelihood

$$\mathbb{P}(v|\theta,B) \propto \exp(\tau_1 \sum_{i,j,k} B_{jik}\langle \theta_i, v_{jk} \rangle).$$

Integrating the latent Beta Process, it can be verified that $B$ follows an IBP marginally [4], i.e. $\mathbb{P}(B) = \mathrm{IBP}(\{m_i\})$.

# 2   Proofs for Lemma 1 and Propositions

## 2.1   Proof for Lemma 1.

**Lemma 1.** $\Gamma : \{\beta = (\beta_1,\ldots,\beta_V) \in \Delta^{V-1} : \beta_i = 0 \text{ for some } i\} \to \{\theta \in \mathbb{S}^{V-1} : \mathbb{1}_V^T \theta = 0\}$ is a homeomorphism, where $\Gamma(\beta) = (\beta - C)/\|\beta - C\|_2$, and $\Gamma^{-1}(\theta) = -\frac{\theta}{\max_i \theta_i/c_i} + C$, for any $C = (c_1,\ldots,c_V) \in \Delta^{V-1}$.

*Proof.* Given any $\beta \in \{\beta = (\beta_1,\ldots,\beta_V) \in \Delta^{V-1} : \beta_i = 0 \text{ for some } i\}$ let $\Gamma(\beta) = (\beta - C)/\|\beta - C\|_2$. Clearly this is a continuous map. Consider the maps $\Gamma_\eta(\theta) = \eta\theta + C$. We show $\Gamma_{\eta_\theta} = \tilde{\Gamma}^{-1}$, for $\eta_\theta = -\frac{1}{\max_i \theta_i/c_i}$. Notice that $\Gamma_{\eta_\theta}(x) \in \Delta^{V-1}$, since $\Gamma_{\eta_\theta}(\theta) = 1$ and $\Gamma_{\eta_\theta}(\theta)_i \geq 0$ for all $i$. The boundary condition $\Gamma_{\eta_\theta}(\theta)_i = 0$ for some $i$ is also satisfied, therefore the range of the map $\Gamma_\eta(\cdot)$ is $\{\beta = (\beta_1,\ldots,\beta_V) \in \Delta^{V-1} : \beta_i = 0 \text{ for some } i\}$, when $\theta \in \{\theta \in \mathbb{S}^{V-1} : \mathbb{1}_V^T \theta = 0\}$. For any $\theta \in \{\theta \in \mathbb{S}^{V-1} : \mathbb{1}_V^T \theta = 0\}$, $\Gamma_{\eta_\theta} \odot \tilde{\Gamma}(\theta) = \frac{\theta}{\|\theta\|_2} = \theta$ as $\|\theta\|_2 = 1$. The right inverse property is proved similarly. $\square$

## 2.2   Proof for Proposition 1.

**Proposition 1.** Given the cost

$$C_{ik}^{(t)} = \begin{cases} \|\tau_1 v_k^{(t)} + \tau_0 \theta_i^{(t-1)}\|_2 - \tau_0 + \log\frac{m_i^{(t-1)}}{t - m_i^{(t-1)}}, i \leq L_{t-1} \\ \tau_1 + \log\frac{\gamma_0}{t} - \log(i - L_{t-1}), L_{t-1} < i \leq L_{t-1} + K^{(t)} \end{cases}$$

consider the optimization problem $\max_{B^{(t)}} \sum_{i,k} B_{ik}^{(t)} C_{ik}^{(t)}$ subject to the constraints that (a) for each fixed $i$, at most one of $B_{ik}^{(t)}$ is 1 and the rest are 0, and (b) for each fixed $k$, exactly one of $B_{ik}^{(t)}$ is 1

and the rest are 0. Then, the MAP estimate for Eq. (1) can be obtained by the Hungarian algorithm, which solves for $((B_{ik}^{(t)}))$ to obtain:

$$\theta_i^{(t)} = \begin{cases} \frac{\tau_1 v_k^{(t)} + \tau_0 \theta_i^{(t-1)}}{\|\tau_1 v_k^{(t)} + \tau_0 \theta_i^{(t-1)}\|_2}, & \text{if } \exists\, k \text{ s.t. } B_{ik}^{(t)} = 1 \text{ and } i \leq L_{t-1} \\ v_k^{(t)}, & \text{if } \exists\, k \text{ s.t. } B_{ik}^{(t)} = 1 \text{ and } i > L_{t-1} \text{ (new topic)} \\ \theta_i^{(t-1)} & \text{otherwise (topic is dormant at } t). \end{cases}$$

*Proof.* First we express the logarithm of the posterior distribution Eq. (1) in a form of a *matching* problem by splitting the terms related to previously observed topics and new topics:

$$\log(\mathbb{P}(\theta^{(t)}, B^{(t)} | \theta^{(t-1)}, v^{(t)}, \{B^{(a)}\}_{a=1}^{t-1})) =$$

$$= \sum_{i=1}^{L_{t-1}} \langle \tau_1 \sum_k B_{ik}^{(t)} v_k^{(t)} + \tau_0 \theta_i^{(t-1)}, \theta_i^{(t)} \rangle + \sum_{i=1}^{L_{t-1}} \sum_k B_{ik}^{(t)} \log \frac{m_i^{(t-1)}}{t - m_i^{(t-1)}} + \tag{7}$$

$$+ \sum_{i=L_{t-1}+1}^{L_{t-1}+K^{(t)}} \tau_1 \langle \sum_k B_{ik}^{(t)} v_k^{(t)}, \theta_i^{(t)} \rangle + \sum_{i=L_{t-1}+1}^{L_{t-1}+K^{(t)}} \sum_k B_{ik}^{(t)} \left( \log \frac{\gamma_0}{t} - \log \frac{(i - L_{t-1})!}{(i - L_{t-1} - 1)!} \right).$$

Next, consider the simultaneous maximization of $B^{(t)}$ and $\theta^{(t)}$. For $i \in \{1, \ldots, L_{t-1}\}$, if $B_{ik}^{(t)} = 1$, i.e., $v_k^{(t)}$ is a noisy version of $\theta_i^{(t)}$, then the increment in the posterior probability is:

$$\log \frac{m_i^{(t-1)}}{t - m_i^{(t-1)}} + \langle \tau_1 v_k^{(t)} + \tau_0 \theta_i^{(t-1)}, \theta_i^{(t)} \rangle.$$

On the other hand, if $B_{ik}^{(t)} = 0$, this increment becomes

$$\log \frac{m_i^{(t-1)}}{t - m_i^{(t-1)}} + \langle \tau_0 \theta_i^{(t-1)}, \theta_i^{(t)} \rangle.$$

Von Mises-Fisher distribution is conjugate to itself and so it admits a closed form MAP estimator:

$$\theta_i^{(t)} = \frac{\tau_1 \sum_k B_{ik}^{(t)} v_k^{(t)} + \tau_0 \theta_i^{(t-1)}}{\|\tau_1 \sum_k B_{ik}^{(t)} v_k^{(t)} + \tau_0 \theta_i^{(t-1)}\|_2}.$$

Plugging this in, the difference between increments defining the cost of the Hungarian objective function is:

$$C_{ik}^{(t)} = \log \frac{m_i^{(t-1)}}{t - m_i^{(t-1)}} + \|\tau_1 v_k^{(t)} + \tau_0 \theta_i^{(t-1)}\|_2 - \tau_0.$$

For $i > L_{t-1}$, it is seen easily from our representation in Eq. (7) and recalling uniform prior for the new global topics, that the reward term of the objective becomes $C_{ik}^{(t)} = \tau_1 + \log(\gamma_0/t) - \log(i - L_{t-1})$ and that given $B_{ik}^{(t)} = 1$, objective function is maximized for $\theta_i^{(t)} \in \mathbb{S}^{V-2}$, when $\theta_i^{(t)} = v_k^{(t)}$. □

## 2.3 Proof for Proposition 2.

**Proposition 2.** Given the cost

$$C_{jik} = \begin{cases} \tau_1 \|v_{jk} + \sum_{-j,i,k} B_{-jik} v_{-jk}\|_2 - \tau_1 \|\sum_{-j,i,k} B_{-jik} v_{-jk}\|_2 + \log \frac{m_{-ji}}{J - m_{-ji}}, & \text{if } i \leq L_{-j} \\ \tau_1 + \log \frac{\gamma_0}{J} - \log(i - L_{-j}), & \text{if } L_{-j} < i \leq L_{-j} + K_j, \end{cases}$$

where $-j$ denotes groups excluding group $j$ and $L_{-j}$ is the number of global topics before group $j$ (due to exchangeability of the IBP, group $j$ can always be considered last). Then, a locally optimum MAP estimate for Eq. (6) can be obtained by iteratively employing the Hungarian algorithm to solve: for each group $j$, $(((B_{jik})))$ which maximizes $\sum_{j,i,k} B_{jik} C_{jik}$, subject to constraints: (a) for each fixed $i$ and $j$, at most one of $B_{jik}$ is 1, rest are 0 and (b) for each fixed $k$ and $j$, exactly one of $B_{jik}$ is 1, rest are 0. After solving for $(((B_{jik})))$, $\theta_i$ is obtained as:

$$\theta_i = \frac{\sum_{j,k} B_{jik} v_{jk}}{\|\sum_{j,k} B_{jik} v_{jk}\|_2}. \tag{8}$$

*Proof.* First, express the logarithm of the posterior probability given in Eq. (6):

$$\log(\mathbb{P}(B, \theta|v)) = \tau_1 \sum_{i,j,k} B_{jik} \langle \theta_i, v_{jk} \rangle + \log \mathrm{IBP}(\{m_i\}) \tag{9}$$

Given $B$, due to vMF conjugacy, MAP estimate of $\theta_i$ is:

$$\theta_i = \frac{\sum_{j,i,k} B_{jik} v_{jk}}{\| \sum_{j,i,k} B_{jik} v_{jk} \|_2} \tag{10}$$

Plugging this into the first part of Eq. (9):

$$\tau_1 \sum_{i,j,k} B_{jik} \langle \frac{\sum_{j,k} B_{jik} v_{jk}}{\| \sum_{j,k} B_{jik} v_{jk} \|_2}, v_{jk} \rangle =$$
$$= \tau_1 \sum_i \| \sum_{j,k} B_{jik} v_{jk} \|_2 = \tau_1 \sum_i \| \sum_k B_{jik} v_{jk} + \sum_{-j,k} B_{-jik} v_{-jk} \|_2. \tag{11}$$

Consider the above objective w.r.t. $\{B_{jik}\}_{i,k}$, i.e. for some fixed $j$, and note that $\sum_k B_{jik} \in \{0, 1\}$. Then equivalent objective function is:

$$\tau_1 \sum_{i,k} B_{jik} \| v_{jk} + \sum_{-j,k} B_{-jik} v_{-jk} \|_2 - \tau_1 \sum_{i,k} B_{jik} \| \sum_{-j,k} B_{-jik} v_{-jk} \|_2. \tag{12}$$

Now consider second term of Eq. (9):

$$\log \mathrm{IBP}(\{m_i\}) = \log \mathbb{P}(\{\sum_k B_{jik}\}_i | \{m_{-ji}\}) + \log \mathrm{IBP}(\{m_{-ji}\}), \tag{13}$$

where $m_{-ji}$ is the number of topics assigned to global topic $i$ outside of group $j$. Due to exchangeability of the IBP, group $j$ can always be considered last and since we optimize for a fixed $j$ given the rest, we can ignore $\log \mathrm{IBP}(\{m_{-ji}\})$.

$$\log \mathbb{P}(\{\sum_k B_{jik}\}_i | \{m_{-ji}\}) =$$
$$= \sum_{i=1}^{L_{-j}} \sum_{k=1}^{K_j} B_{jik} \log \frac{m_{-ji}}{J - m_{-ji}} + \sum_{i=L_{-j}+1}^{L_{-j}+K_j} \sum_{k=1}^{K_j} B_{jik} \left( \log \frac{\gamma_0}{J} - \log(i - L_{-j}) \right). \tag{14}$$

Finally observe that when $i > L_{-j}$, Eq. (12) reduces to $\tau_1$. Combining this observation, Eq. (12) and Eq. (14) we arrive at the desired cost formulation. $\square$

## 3 Algorithms description

**Streaming Dynamic Matching.** SDM algorithm is described in Section 3 of the main text. Obtaining topic estimates with CoSAC [6] for different time steps can be performed in parallel if data is available in advance as it was done in the Wiki experiments for SDM with 20 cores.

**Distributed Matching.** Using CoSAC, for groups $j = 1, \ldots, J$, we obtain topics $\{\beta_{jk} \in \Delta^{V-1}\}_{k=1}^{K_j}$ from $\{x_{jm} \in \mathbb{N}^V\}_{m=1}^{M_j}$, collection of $M_j$ documents of group $j$. The above step can trivially be done in parallel. Estimated topics are then transformed to $\{v_{jk} \in \mathbb{S}^{V-2}\}_{k=1}^{K_j}$ using Lemma 1 and reference point $C = \sum_{j=1}^{J} \sum_{m=1}^{M_j} \frac{x_{jm}}{N_{jm}} / \sum_{j=1}^{J} M_j$, where $N_{jm}$ is the number of words in the corresponding document. This reference point is simply a mean of the normalized documents across groups. Finally we compute MAP estimates of global topics by iterating over groups and updating assignments based on Proposition 2. Distributed Matching (DM) is summarized in Algorithm 1. For initializing the algorithm we can use SDM over random sequence of groups.

---
**Algorithm 1** Distributed Matching (DM)

---
1: **for** $j = 1, \ldots, J$ (in parallel) **do**
2:     Estimate topics $\{\beta_{jk}\}_{k=1}^{K_j} = \text{CoSAC}(\{x_{jm}\}_{m=1}^{M_j})$
3:     Map topics to sphere $\{v_{jk}\}_{k=1}^{K_j}$ (Lemma 1)
4: **end for**
5: **repeat**
6:     select random group index $j$
7:     Given $\{v_{jk}\}_{jk}$ update $(((B_{jik})))$ for group $j$ using Proposition 2
8: **until** convergence
9: Obtain MAP estimates of $\{\theta_i\}_i$ given $(((B_{jik})))$

---

**Streaming Dynamic Distributed Matching.** SDDM is a synthesis of our SDM and DM algorithms. At time $t$, using CoSAC, for groups $j = 1, \ldots, J$, we obtain topics $\{\beta_{jk}^{(t)} \in \Delta^{V-1}\}_{k=1}^{K_j^{(t)}}$ from $\{x_{jm}^{(t)} \in \mathbb{N}^V\}_{m=1}^{M_j^{(t)}}$, collection of $M_j^{(t)}$ documents of group $j$ at time $t$. The above step is done in parallel. Estimated topics are then transformed to $\{v_{jk}^{(t)} \in \mathbb{S}^{V-2}\}_{k=1}^{K_j^{(t)}}$ using Lemma 1 and reference point $C_t = \sum_{a=1}^{t} \sum_{j=1}^{J} \sum_{m=1}^{M_j^{(a)}} \frac{x_{jm}^{(a)}}{N_{jm}^{(a)}} / \sum_{a=1}^{t} \sum_{j=1}^{J} M_j^{(a)}$, where $N_{jm}^{(a)}$ is the number of words in the corresponding document. Then we update estimates of global topics dynamics based on the results of Section 3.3 of the main text. Streaming Dynamic Distributed Matching (SDDM) is summarized in Algorithm 2.

---
**Algorithm 2** SDDM

---
1: **for** $t = 1, \ldots, T$ **do**
2:     **for** $j = 1, \ldots, J$ (in parallel) **do**
3:         Estimate topics $\{\beta_{jk}^{(t)}\}_{k=1}^{K_j^{(t)}} = \text{CoSAC}(\{x_{jm}^{(t)}\}_{m=1}^{M_j^{(t)}})$
4:         Map topics to sphere $\{v_{jk}^{(t)}\}_{k=1}^{K_j^{(t)}}$ (Lemma 1)
5:     **end for**
6:     **repeat**
7:         select random group index $j$
8:         Given $\{v_{jk}^{(t)}\}_{jk}$ and $\{\theta_i^{(t-1)}\}_i$ update $(((B_{jik}^{(t)})))$ for group $j$ using cost defined in Section 3.3 of the main text
9:     **until** convergence
10:     Obtain MAP estimates of $\{\theta_i^{(t)}\}_i$ given $(((B_{jik}^{(t)})))$ and $\{\theta_i^{(t-1)}\}_i$
11: **end for**

---

**Remark** Each time we apply Hungarian algorithm, we can at most discover $K$ topics, where $K$ is the number of local topics used for constructing the cost. It is possible to control the growth of the number of topics by setting a saturation value. When number of global topics exceeds saturation value, we can allow only limited amount of topics to be added each time we apply Hungarian algorithm by truncating cost to $i \leq L + c$ (instead of $i \leq L + K$), where $L$ is the number of global topics and $c$ is the maximum number of topics that can be added when $L$ exceeds the saturation value. When exploring parameter sensitivity we set saturation value to 250 and $c = 1$. Our algorithms remain nonparametric, however this helps to avoid excessively large number of global topics for extreme cases of parameter values.

Another empirical modification we found useful, especially on the EJC data with large number of groups, is to truncate popularity counts $m$ at some value (we used 10) when constructing the cost. This helps to prevent extreme rich-get-richer behavior, i.e. when $\log \frac{m_i}{J - m_i}$ becomes too large for some popular topic $i$ and this topic continues to be assigned to even when there is potentially a better match.

|  (a) EJC perplexity | (b) # of topics in EJC | (c) Wiki perplexity | (d) # of topics in Wiki |

Figure 1: Perplexity and number of topics of SDM

# 4 Experiments details

Here we provide some additional details and hyperparameter settings for the experiments. The Dynamic Topic Model [2] was trained using code from `https://github.com/blei-lab/dtm` with default parameter settings and $K = 100$. In 56 hours we were able to complete LDA initialization and two EM iterations of the dynamic updates. Streaming Variational Bayes [3] was trained based on the code from `https://github.com/tbroderick/streaming_vb` with default parameters, $K = 100$, $\eta = 0.01$, batch size of 2048 and 4 asynchronous batches per evaluation. For CoSAC [6], we used code from `https://github.com/moonfolk/Geometric-Topic-Modeling` with default parameters and $\omega = 0.7$. Same setting of CoSAC was used for learning topics on batches in our framework.

To compare perplexity of all algorithms we utilized geometric approach for computing document topic proportions *given* topics from Yurochkin & Nguyen [5]. Code for this procedure is available at `https://github.com/moonfolk/Geometric-Topic-Modeling`.

# 5 Extended sensitivity results

We present additional results regarding sensitivity of hyperparameters of SDM in Fig. 1. It is interesting to see the impact of large $\tau_0$. This parameter influences the variability of global topics across time and when this variability is small, global topics are reluctant to changes, hence more global topics needed to fit the local topics. We also note that smaller $\tau_0$ results in better perplexity on the EJC corpora, while having little effect on the perplexity of the Wiki dataset. EJC topics appear to be changing at a faster rate, while Wiki topics are relatively more stable.

# 6 Relative comparison to other methods

For the Wiki corpus we considered Fast DTM [1], but the implementation available online appeared not efficient enough (multicore implementation of Fast DTM is not published). We report a relative comparison to our results based on the best run-time reported in their work. We also note that approach of Bhadury et al. [1] is not suitable for streaming since computations are parallelized across time slices. Additionally, Streaming Variational Bayes (SVB) [3] appeared quite slow on our computing cluster, therefore we consider it for the relative comparison as well. In Table 1 we compare best run-times reported in the respective papers with the run-time of SDDM. Our method can be seen to be significantly faster.

Table 1: Running time comparison

|  | Data size | Training Time | Cores used |
| --- | --- | --- | --- |
| SDDM | 3mil | 20min | 20 |
| SVB | 3.6mil | 125min | 32 |
| Fast DTM | 2.6mil | 28min | 58 |

# 7  Datasets

## 7.1  Early Journal Content

For the EJC dataset we get from `http://www.jstor.org`, the data is well-structured and the preprocessed 1-gram format is available along with the corresponding meta data in xml format for each document. We performed stemming on every word shown in the dataset and removed all stop words given in ENGLISH_STOP_WORDS from `sklearn.feature_extraction.stop_words` and `nltk.corpus.stopwords.words`. Words the length of which are shorter than 3 are removed. We also removed those words that appear in more than $99\%$ of the documents and those appearing in less than $1\%$ of the documents. For documents, we removed those *outliers*, in which a same word appears more than 200 times. Those documents which contain less than 100 words (considering only words in the preprocessed vocabulary) are also excluded. After preprocessing, there are 4516 words in the vocabulary and approximately $400k$ documents left. We batch the documents based on both of their publication years and the journals they are published in.

## 7.2  Wikipedia data

For Wikipedia data, we need four main components: *the vocabulary*, *the original Wikipedia page texts*, *the page view counts* (for each Wikipedia page being considered), and the *category-title mapping*. The data acquisition and processing for each component is described separately as follows.

**The vocabulary**  We take the vocabulary from Wiktionary:Frequency_lists Originally there are 10000 words. We removed words shorter than 3 characters and removed all stop words given in ENGLISH_STOP_WORDS from `sklearn.feature_extraction.stop_words` and `nltk.corpus.stopwords.words` After preprocessing, there are 7359 words left in the vocabulary.

**The original Wikipedia page texts**  We downloaded the Wiki data dumps (2017/08/20) from `https://meta.wikimedia.org/wiki/Data_dump_torrents#English_Wikipedia` containing about 9 million Wikipedia pages after decompression. We split the whole file into multiple text files in which each individual file contains the content of a single Wikipedia page. We then use *MeTA*, a modern C++ data sciences toolkit to transform all of these raw texts files into 1-gram format using the preprocessed vocabulary.

**The page view counts**  We use the Pageview API provided by AQS (Analytics Query Service) to get the page view count information of each Wikipedia page by specifying the time period of interest (year 2017) and the granularity (monthly). The API will return the page view count information as a json file in which the page view count is given for every month in the year 2017.

**The category-title mapping**  Wikipedia pages are categorized in a structure called *category tree*. There are 22 *top-level* categories (e.g., Arts, Culture, Events, etc). Under each category, there could be relevant Wikipedia pages and subcategories, and each subcategory could also contain another set of subcategories and corresponding pages. Unlike the name suggests, *category tree* is *cyclic* and is in fact not tree-structured: each category could be under multiple categories and each page could belong to multiple categories/subcategories and there could be *cycles*. Thus trying to traverse the whole tree to get articles under a certain category is infeasible. Considering all the categories/subcategories in the category tree could also be distractive. Thus, we only focus on the *top-level* categories and exclude the category *Reference works* since it is of little interest. We use *wptools* to traverse the category tree for each of the top-level categories of interest *individually* by specifying the category name during the traversal. We store the mapping information between each category and the corresponding Wikipage titles for later processing.

**Components aggregation**  We first perform *intersection* over the titles of Wikipedia pages extracted from the original Wikipedia page texts, page view counts and category-title mapping, and drop those Wikipedia pages the texts in which contain less than 10 words (the total word count) from the vocabulary. Since there is a severe inconsistency in titles extracted from all the three components, performing intersection results in loss of a large portion of Wikipedia pages. After dropping all unqualified Wikipedia pages, we have approximately 500k remaining Wikipedia pages. We assign

each Wikipedia page a timestamp based on its page view counts across the 12 months in 2017, the month in which the page gets most view counts is assigned to the page as its timestamp. Since our model naturally considers data in *two-level* batches (for example, we batch the Wikipedia pages based on time and category), and there are overlapping Wikipedia pages among different batches (one Wikipedia page could belong to multiple categories), we have about three million Wikipedia pages in the final dataset incarnation across all the batches.