[Reviews · NeurIPS 2019]

Reviewer 1



The authors propose a set of models and inference algorithms to model the evolution of topics over time. The proposed set of models are novel in terms of both the generative models and inference techniques. The novelty in generative models is achieved by representing a set of topics at a given time as a topic polytope and modeling evolution of topics as trajectories in the geometric space of the polytopes. The proposed inference approach is fast and scalable, and lends itself well to online learning and distributed computing. The authors describe the motivation for the generative process and the algorithms in sufficient detail. The supplement further elaborates on inference details and sensitivity to priors. Experimental results demonstrate improvements in perplexity and significant speed ups in run time. The major contributions are clear and significantly advance the state of research.

Reviewer 2



Originality: This is a nice combination using the vMF model and Hungarian matching. I really like the ideas here and the performance is great Quality and Clarity: The basics are ok: citing related work, using those ideas to make your point, etc. But I feel that you miss some opportunities to make the work more accessible. You gave Algorithm 1, but it is hardly worth it, I needed more detail. Also, if you are going to give an algorithm block, why not for SDDM? Please clearly link the mathematical ideas to the algorithm? Likewise, the math jumps in hard to follow ways. You may laugh but I am not sure where the cost (164, 214 and 244) are coming from and how the ideas are represented in those formulas. The body of the paper reference nice mathematical ideas but I found it difficult to get the ideas. Significance: as I said above, the speed of this approach seems very valuable.

Reviewer 3



This is a very well written paper, both in style and substance. There are a few stylistic peculiarities that could surely be ruled out by thorough proof-reading. The authors present a nice introduction into the idea of modelling sets of topics, i.e. sets of points on a simplex, as the geometric structure of a polytope. They go on to describe, how evolution of such a polytope can be modelled over time by embedding a unit hypersphere into the simplex and modelling polytope evolution as random trajectories over this sphere. They further present a non-parametric hierarchical model for capturing polytopes with a varying number of topics and also multiple polytopes arising from different corpora. Their experimental section deals with two different data sets, a medium sized one (400k documents) and a large one (3M documents). While the sizes of the corpora are appropriate to demonstrate the performance of the inference algorithm, usage of more well-known corpora such as the NYT corpus would have been beneficial in terms of comparability to previous approaches to the problem. Also, vocabulary truncation to just over 4500 terms from 400k documents and 7300 words from 3M documents seems rather aggressive and needs further elaboration. Although there is a case study for a certain topic, including probability trajectories for its top words, a more extensive qualitative assessment of model performance would be beneficial. Since topic models are primarily successful because of their interpretability by humans, it is often useful to demonstrate qualitative over quantitative results.

[Author Response · NeurIPS 2019]

We thank all the reviewers for their time, valuable and encouraging feedback and recommendations for improvement.
Answers to specific comments appear below.

***R1:*** *"While it is clear that SDM and SDDM generally outperform other models in terms of perplexity, the authors should*
*comment more on ideal settings for each approach as there is no clear winner between the two based on experimental*
*results."*

SDDM should be superior to SDM when the time-group partition of the data allows for sufficient number of documents
per time-group slice to learn local topics. On the EJC dataset, majority of the groups have very few articles leading to
lower quality local topics. This observation is briefly summarized in lines 301-302 of the main text - we will emphasize
this point more clearly in the revised version.

***R2's*** *comments regarding clarity of the presentation*

We appreciate the suggestions for improving the clarity of the paper. If the paper is accepted, the 9th content page
(allowed for the camera-ready version) will help us to incorporate these suggestions and improve the flow of the paper.

***R3:*** *"Elaborate on data set preparation"*

Regarding vocabulary sizes, for the Wikipedia corpus, we followed similar procedure for vocabulary truncation as
in Online Learning for Latent Dirichlet Allocation (Hoffman et al., 2010; they also analyzed over 3mil Wikipedia
articles and truncated vocabulary to 7995 words as stated in their footnote 4). We describe the Wikipedia vocabulary
preparation in lines 146-150 of the Supplement and we will add the reference to clarify our choice of the vocabulary.
On the EJC dataset we applied relatively standard vocabulary truncation steps, i.e. removing very common (over 99%
documents) and rare (under 1% documents) words, removing short and stop words, and stemming. This procedure is
described in Supplement section 7.1. To argue that the resulting vocabulary size of 4516 words is appropriate, we may
compare to the vocabulary size of 4253 words used by Hoffman et al. (2010) on the 350k documents Nature corpus (see
footnote 3 in their paper; we also note that Nature corpus is not public leading to us choosing the EJC corpus instead).

***R3:*** *"Extend qualitative analysis. For this, the quantitative analysis of hyperparameter influence could be pushed into*
*the supplemental."*

We thank the reviewer for the suggestion and we will add additional qualitative examples in the revised version.

[Meta-Review · NeurIPS 2019]

The three reviewers all agreed during discussion that the paper should be accepted. They "would appreciate if the authors incorporate the suggestions provided. It would strengthen the paper and make it more accessible to the wider research community." They also hope that the suggestions are not "taken superficially by the authors. The paper would be much better if the discussion, math and especially the algorithms were more clearly linked."